# The morphospace of the brain-cognition organisation

Valentina Pacella ●[1,2,3] ✉, Victor Nozais ●[2,3], Lia Talozzi ●[2,3,4], Majd Abdallah[2,3,5,6], Demian Wassermann ●[5,6], Stephanie J. Forkel[3,7,8,9] & Michel Thiebaut de Schotten ●[2,3] ✉

Over the past three decades, functional neuroimaging has amassed abundant evidence of the intricate interplay between brain structure and function. However, the potential anatomical and experimental overlap, independence, granularity, and gaps between functions remain poorly understood. Here, we show the latent structure of the current brain-cognition knowledge and its organisation. Our approach utilises the most comprehensive meta-analytic fMRI database (Neurosynth) to compute a three-dimensional embedding space–morphospace capturing the relationship between brain functions as we currently understand them. The space structure enables us to statistically test the relationship between functions expressed as the degree to which the characteristics of each functional map can be anticipated based on its similarities with others–the predictability index. The morphospace can also predict the activation pattern of new, unseen functions and decode thoughts and inner states during movie watching. The framework defined by the morphospace will spur the investigation of novel functions and guide the exploration of the fabric of human cognition.

Functional neuroimaging has provided substantial evidence of the intricate relationship between the brain's structure and function. Functional MRI (fMRI) studies explored this relationship by identifying specific clusters of neural activations corresponding to cognitive domains such as motor, auditory, and visual functions. However, other cognitive functions may not as clearly segregate into brain clusters as demonstrated by overlapping activation patterns[1] observed for working memory[2], spatial attention[3,4], language[5], and mirror neurons[6]. The fact that our current understanding of cognitive functions is not entirely segregated challenges the conventional brain-cognition mapping built upon activation models. This is because our representation of cognition is largely based on theoretical and experimental paradigms that are recursively validated. Hence, while functional neuroimaging has revolutionised our understanding of the brain's structure-function relationship, it has also exposed limitations in our ability to formulate coherent theories of cognition.

Cognitive neuroscience relies on evolving concepts linking our mind to our brain[7]. Brain mapping evolved from simple one-to-one correlations to complex neural systems driving human behaviour[8,9]. However, the vast increase of task-related fMRI studies coupled with the replication crisis has hampered progress in the theorisation of cognitive concepts[10,11], resulting in insufficient knowledge of the potential overlap, independence, granularity, and gaps between functions at the global level.

[1]IUSS Cognitive Neuroscience (ICON) Center, Scuola Universitaria Superiore IUSS, Pavia, Italy. [2]Groupe d'Imagerie Neurofonctionnelle, Institut des Maladies Neurodégénératives-UMR 5293, CNRS, CEA, University of Bordeaux, Bordeaux, France. [3]Brain Connectivity and Behaviour Laboratory, Paris, France. [4]Department of Neurology and Neurological Sciences, Stanford University School of Medicine, Stanford, CA, USA. [5]MIND team, Inria Saclay Île-de-France, Université Paris-Saclay, 1 Rue Honoré d'Estienne d'Orves, Palaiseau, Ile-de-France, France. [6]Neurospin, CEA, Gif-sur-Yvette, Ile-de-France, France. [7]Donders Centre for Brain Cognition and Behaviour, Radboud University, Thomas van Aquinostraat 4, Nijmegen, the Netherlands. [8]Centre for Neuroimaging Sciences, Department of Neuroimaging, Institute of Psychiatry, Psychology and Neuroscience, King's College London, London, United Kingdom. [9]Max Planck Institute for Psycholinguistics, 6525 XD Nijmegen, Wundtlaan 1, the Netherlands. ✉e-mail: valentina.pacella.90@gmail.com; michel.thiebaut@gmail.com

Assessing the degree of coherence in the foundations of cognitive theoretical constructs is crucial to deepen our understanding in the field. Such an attempt is now possible thanks to the recent surge in large dataset studies and the adoption of meta-analytic techniques that have enabled researchers to examine the consistency of neuroimaging findings[12–16] and reduce the likelihood of false positive errors[12]. Advances in dimensionality embedding have provided the means of visualising large data complexity by compressing their variability in a few dimensions, allowing for a more comprehensive investigation of neural cognitive systems as an integrated whole[17–21]. While previous efforts demonstrated that a few embedding components can describe underlying patterns in functional activations[22,23], the wealth of knowledge, data, and publications produced in the field of fMRI has not yet been comprehensively understood within a brain-cognition framework[10,11]. As we accumulate more data elucidating the anatomical foundations of cognition, synthesising these findings into cohesive theories becomes increasingly challenging. To address this complexity, we introduce a 'Morphospace' and a 'predictability index,' which together characterise and quantify the interconnections among task-specific fMRI studies. The predictability index also serves as a metric for assessing the predictive validity of past, present, and future fMRI investigations into cognitive processes. These tools not only help to synthesise existing data but also provide a framework for future research, as evidenced by this proof-of-concept.

## Results

### The brain-cognition organisation framework and the predictability index

In order to extract and reveal the predictability of the current brain-cognition knowledge, we utilised Neurosynth (neurosynth.org), which conducts coordinate-based meta-analyses of fMRI activations associated with specific cognitive terms. We carefully selected the most comprehensive meta-analytic fMRI database of cognition consisting of 506 meta-analytic maps[24]. Each meta-analytic map was parcellated using a multi-atlas approach to create a comprehensive anatomical representation of the meta-analytic activation maps. These parcellated maps were then embedded using the Uniform Manifold Approximation and Projection method (UMAP[25]). UMAP estimated the similarity between the meta-analytic activation maps and represented it as the Euclidean distance between maps, effectively defining the morphospace (Figs. 1A, B). We leveraged the Euclidean distances between the maps in a range of embedding dimensions, from 2 to 5, as independent variables to predict the morphology of each map in the morphospace. The 3-dimensional embedding demonstrated better predictions ($z = 4.37$, $p$-value $< .001$; Fig. 1C), thus better capturing the shared variability between the maps. A neuron-shaped 3D architecture proved to be stable across different parcellation approaches and UMAP features (Materials and Methods and Supplementary Figs. S1 and S2). The space clustered specific cognitive domains along different branches

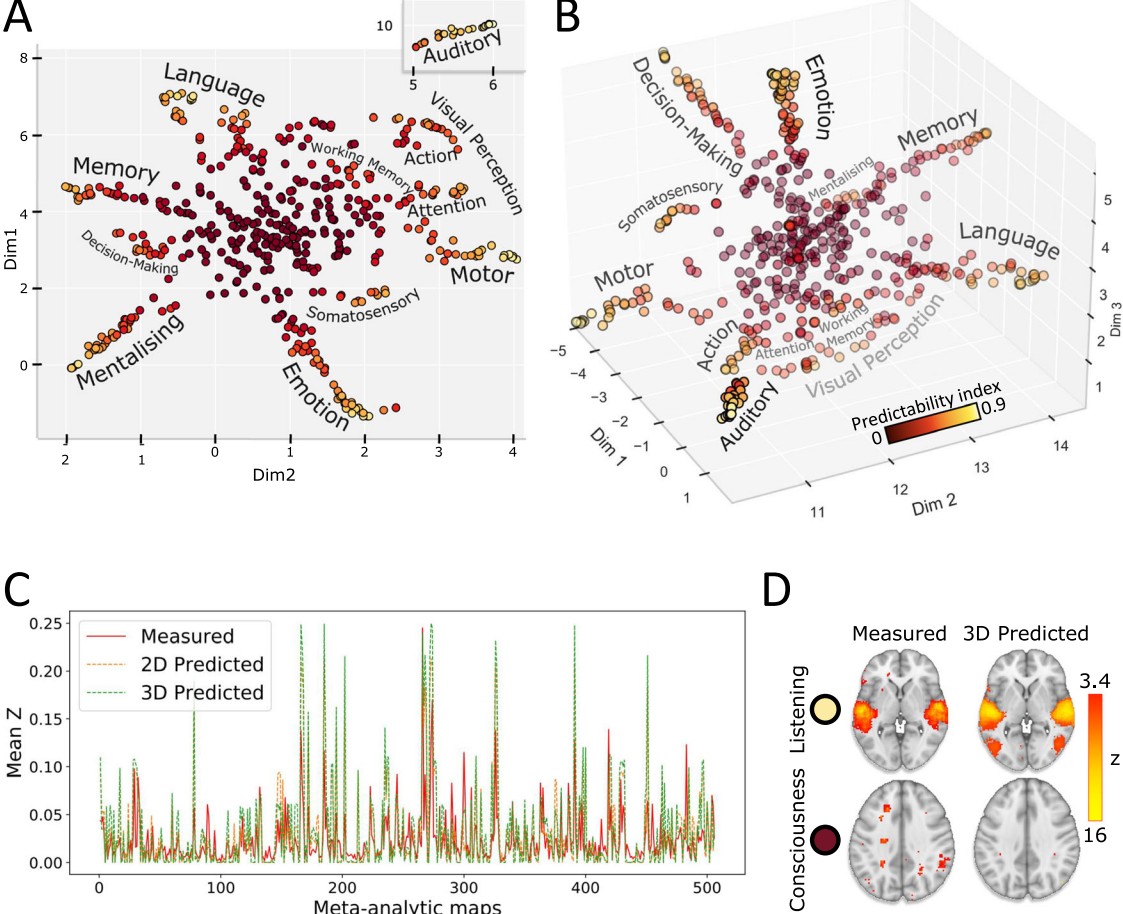

**Fig. 1 | The morphospace and the predictability index.** In the 2D (**A**) and 3D (**B**) morphospace, similar functional meta-analytic maps cluster together. The colour bar indicates the predictability index. Cognitive domains are indicated next to each branch. **C** Comparison between the measured maps and the 2D and 3D predicted maps' mean z (see "Materials and Methods" for 4D, 5D comparison). **D** Representative examples of the best (listening) and worst (consciousness) measured (left) and predicted (right) pair of meta-analytic maps. The colour bar represents the z-statistic of voxels from the Neurosynth meta-analysis maps (measured) and voxels resulting from the voxel-wise linear regression (predicted). Source data are provided as a Source Data file.

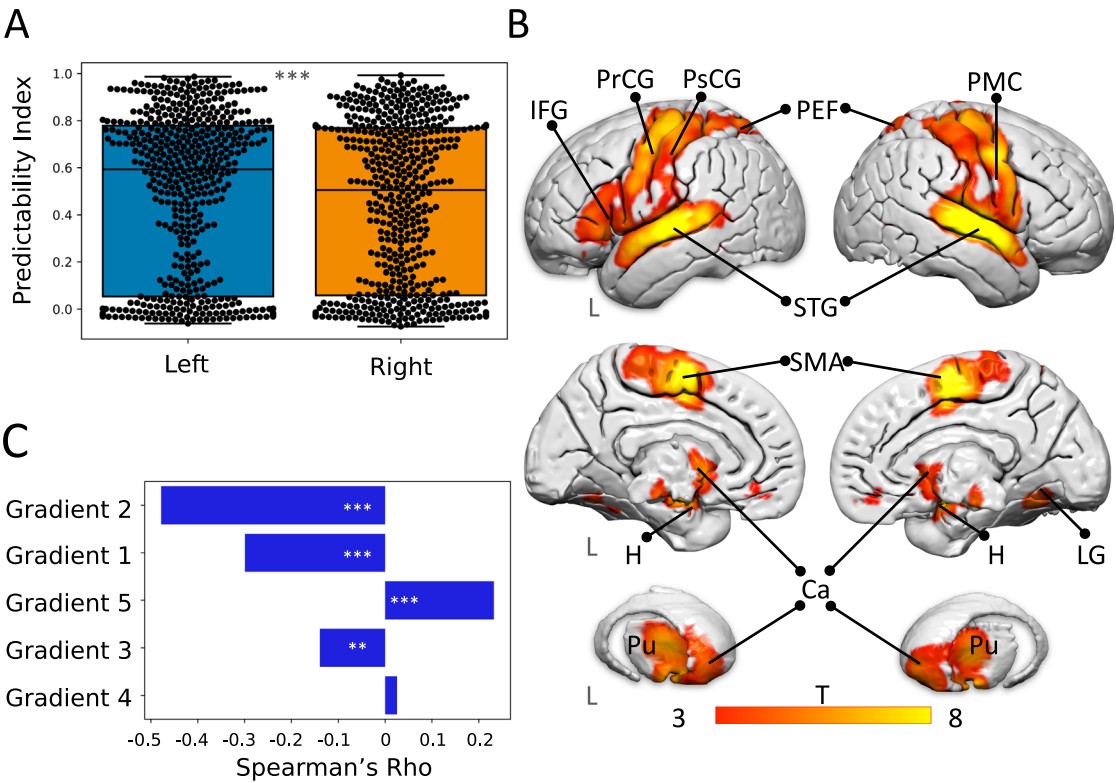

**Fig. 2 | Hemispheric and regional reliability of functional characterisation and their relationship with the data-driven domain-wise organisation of brain functions (i.e., resting-state fMRI gradients). A** The two-sided t-test shows the difference between the mean predictability index of the left (blue) and right (orange) hemispheres. ***, $p < 0.001$. Lines, boxes, whiskers and dots represent the median, quartiles, distribution, and observations (506 meta-analytic maps). **B** Left and right lateral (top), left and right medial (bottom) views and 3D reconstruction of the basal ganglia indicating the brain structures that are the most reliably characterised functionally (predictability map). The colour bar represents the t-statistics. **C** Two-sided Spearman correlation between the predictability map and the five resting-state fMRI gradients. *** = $p < .001$, ** = $p = .004$. Ca: caudate nucleus. H: hippocampus. IFG: inferior frontal gyrus. L: left hemisphere. LG: lingual gyrus. PEF: parietal eye field. PMC: premotor cortex. PrCG: precentral gyrus. PsCG: post-central gyrus. Pu: putamen. SMA: supplementary motor area. STG: superior temporal gyrus. Source data are provided as a Source Data file.

(e.g., attention, motor, language), while the centre of the space gathered heterogeneous cognitive functions (e.g., consciousness, executive functions).

The morphospace, which embodies our current understanding of the relationship between functions, integrates the theoretical and methodological advances and limitations of three decades of fMRI research. To evaluate the coherence of the organisation of brain functions, we leveraged the spatial properties of the morphospace. This statistical assessment involves comparing new observations with predictions based on a corpus of established, coherent findings[26]. The divergence between these predictions and the actual observation is quantified, offering a metric to gauge and potentially refine the predictability of future estimations[27–29]. This approach is expressed by the predictability index, which utilises linear regression to assess the degree to which the characteristics of each functional map can be anticipated based on its similarities with others measured through tridimensional Euclidean distance. Thus, the predictability index is derived by quantifying the correlation between the predicted and observed maps reflecting the predictability power of each map. Meta-analytic maps contributing to the coherence of the organisation of brain functions had higher predictability indices, as depicted in Fig. 1D. Notably, the less predictable maps were primarily situated in the centre of the morphospace, while the 'neuron-shaped' branches contained more predictable maps. These findings suggest that some cognitive functions may be less coherent in terms of theoretical and methodological scaffolding compared to more demarcated functions.

Moreover, the exploration of functions is also subjected to an anatomical bias. Figure 2A demonstrates the presence of an imbalance between the two hemispheres of the human brain in terms of reliability of functional characterisation (i.e., epistemological imbalance): the left hemisphere shows significantly higher predictability indices compared to the right (t = 3.3, df = 505, p-value < .001). Further regression analysis between the measured meta-analytic maps and their predictability index revealed the brain regions associated with the most accurately predicted functions. These regions included motor, auditory, and primary somatosensory cortices, the medial temporal (MT) cortex, premotor cortex, frontal and parietal eye fields, supplementary motor area (SMA), caudate, putamen, pallidum, substantia nigra, red nucleus and basal forebrain in both hemispheres and posterior inferior frontal cortex (also known as 'Broca's area') in the left hemisphere (Fig. 2B). These findings suggest that the function of these brain areas is more reliably characterised.

**Validation of the predictive framework**

To better understand these regions, the map of the highly characterised structures was compared to five task-free resting-state fMRI gradients, typically employed to describe brain regions along a spectrum going from unimodal to higher-order cognitive functions[22]. The findings revealed a significant negative correlation with the first and second gradients (rho = −0.3 and rho = −0.48, both with p-values < .001; Fig. 2C and a positive correlation with the fifth gradient (rho = 0.23, p-value < .001; Fig. 2C. This suggests that the functionally best-characterised cortical areas are associated with negative values on the

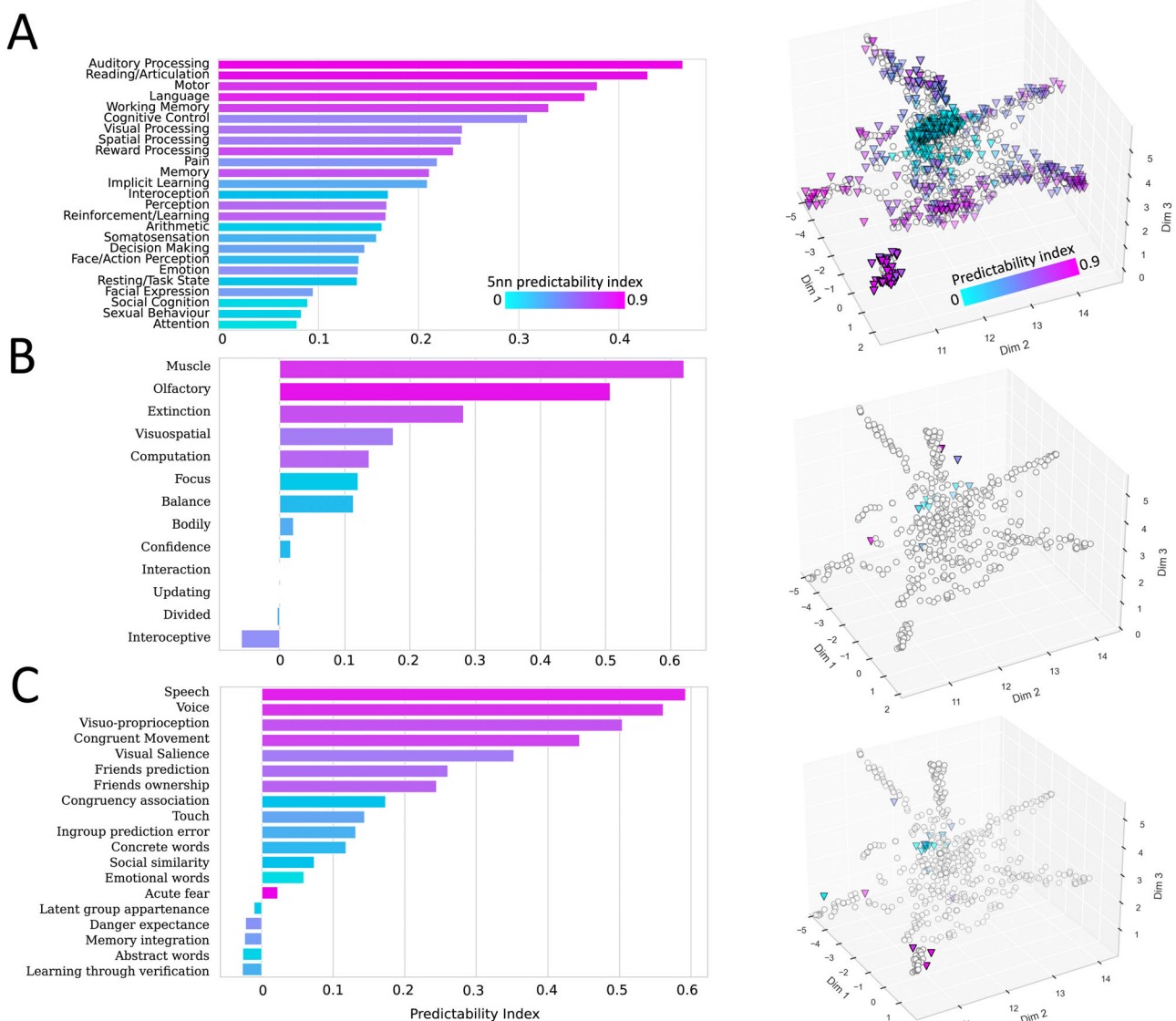

**Fig. 3 | The morphospace ability to predict unexplored functions.** The bar plots displayed on the left indicate the predictability index for (**A**) the Neuroquery meta-analytic maps, **B** Neurosynth meta-analytic maps reported after 2017, and (**C**) independent fMRI maps. These indices were calculated using Spearman rank correlation between predicted and measured activations. The colour for each new map's bar plot corresponds to its five nearest neighbours' (5nn) predictability index. The 888 new maps from Neuroquery (**C**) were summarised into 25 cognitive macro-categories via topic modelling (see "Materials and Methods"). The macro-categories are based on the terms most frequently associated in the literature with the 888 terms of interest. On the right panels, triangles indicate each new map's coordinate in the morphospace, while transparent circles indicate the morphospace meta-analytic maps' location. The "cool" palette characterises the new projected meta-analytic and single activation maps. Source data are provided as a Source Data file.

first two gradients that are mostly representative of unimodal brain systems. Some exceptions, such as the SMA, inferior frontal gyrus ('Broca's area'), and the frontal and parietal eye fields correlated positively with the fifth gradient, reflecting a good characterisation. Thus, the characterisation of high-order systems that process inner state (e.g., consciousness, awareness, mood), which would typically load on the positive values of the first and the second gradients[22], would require a more systematic and rational investigation of the unexplored gaps in the morphospace.

To determine whether the morphospace can provide a rational framework to fill these gaps, we attempted to predict the meta-analytic activation maps of missing (i.e., undetermined) functions. As a proof of concept, unexplored functions in the aforementioned gaps were validated with out-of-sample maps, including meta-analytic maps of new terms derived from an independent meta-analytic repository

(Neuroquery, https://neuroquery.org/; $n = 888$; Fig. 3A), new meta-analytic activation maps derived from Neurosynth after 2017 ($n = 13$; Fig. 3B), and recent independent fMRI results from individual studies released after 2017 (https://neurovault.org/; $n = 19$; see Supplementary Table S1 for a description; see Fig. 3C). Accordingly, a total of $n = 920$ new meta-analytic and activation maps were projected into the morphospace. All new maps landed mostly on the branches and empty locations of the morphospace (Fig. 3). As previously mentioned, we statistically compared these new maps with their predictions. The results indicate that new maps landing close to rationally explored functions were reliably predicted, ranging from low to high effect sizes ($0.1 < r < 0.5$). Hence, the morphospace can be employed to predict the pattern of activation of new and currently undetermined functions as long as they can be logically deduced from our understanding of other functions.

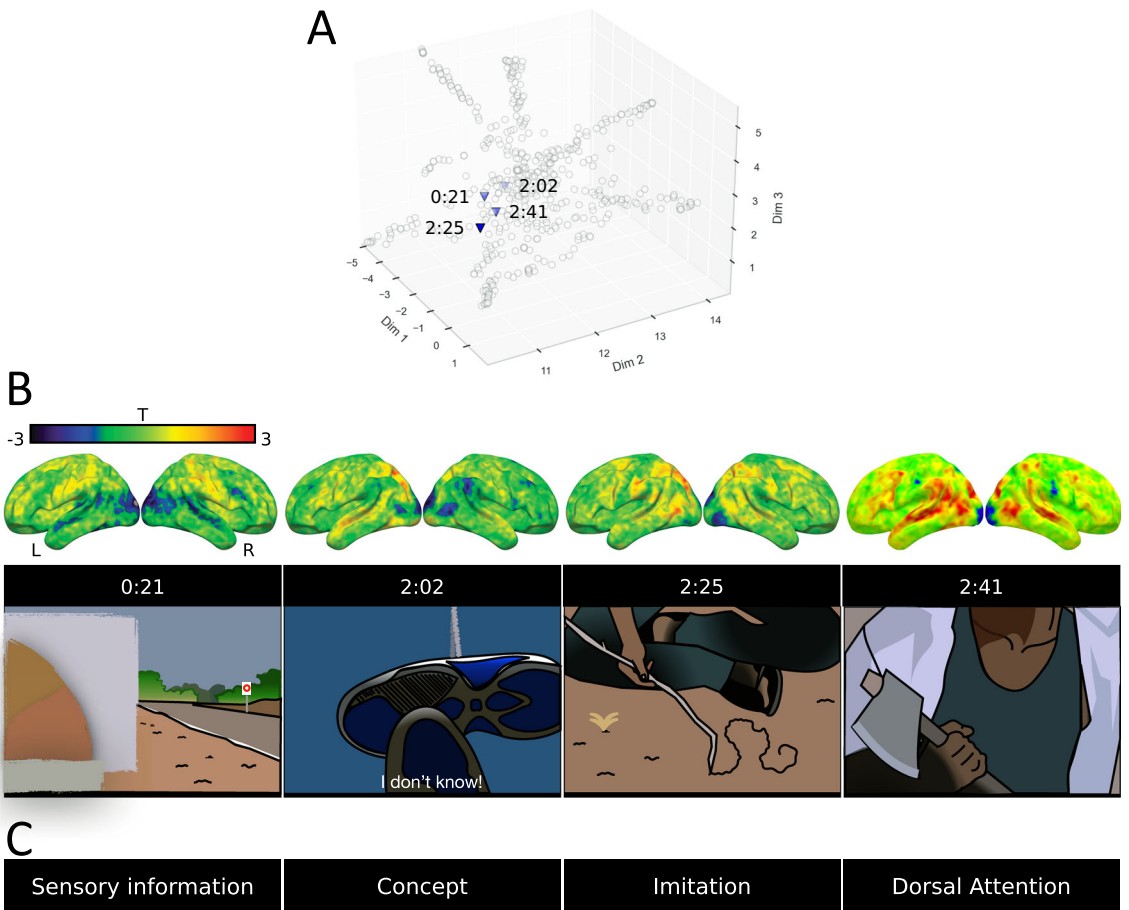

**Fig. 4 | Illustration of the cognitive decoding of 4 representative neural activations during movie observation. A** The projections of the activation maps in the morphospace, triangles indicate each representative map's coordinate in the morphospace, and transparent circles indicate the morphospace meta-analytic maps' location. **B** Localisation of the activations on a brain template (top) and drawings representing the frames from the movie 'Two Man' eliciting the activations (bottom). **C** The cognitive terms whose meta-analytic map had the shortest Euclidean distance from each activation. The colour bar indicates the activations' t-statistics. Source data are provided as a Source Data file.

## Thoughts and inner states anatomical characterisation via the predictive framework

To evaluate the morphospace's capacity to predict inner state-related cognition (i.e., not driven by task execution), we endeavoured to decode neural activations during movie watching. We produced activation maps ($n = 3655$) representing changes in cognitive states at every second in the Human Connectome Project movie dataset (https://www.humanconnectome.org/[30]) and projected the maps onto the morphospace. Our findings demonstrate that the morphospace has the potential to decode participants' thoughts and mental states on a second-by-second basis (Fig. 4).

## Discussion

We introduced a framework for understanding the brain's organisation of cognition, the morphospace, which was derived from an extensive meta-analytic database of task-related functional MRI (fMRI). Our results demonstrate that some functions, particularly those involving primary cortices (e.g., precentral gyrus, occipital cortex) and transmodal areas (e.g., SMA, inferior frontal gyrus), have been more distinctly defined within this morphospace, facilitating their segregation and characterisation. This morphospace not only allows for the predictive modelling of functions that are not yet fully understood, corroborated by external neuroimaging data, but also offers a lens for interpreting cognitive states during task-based and resting states. Overall, our framework promises to streamline the exploration of cognition, enabling more targeted and hypothesis-driven research.

The morphospace represents a neuron-shaped architecture that clusters cognitive domains within each branch, which may suggest a fractal relationship between brain structures and functions[31]. The position of cognitive domains on each branch reflects their interrelationship within and between branches (see Supplementary Materials for an extensive discussion). For instance, the decision-making branch contains meta-analytic activation maps for reward, learning, and prediction. The proximity of domain branches, such as emotion and decision-making, reflects their close theoretical interaction[32]. By contrast, the anatomical pattern of auditory-modality fMRI tasks is strikingly different and requires further investigation into the possible influence of stimuli modality on activation studies. Accordingly, the pronounced distance of the auditory cluster could reflect a fundamental difference with other cognitive domains that are systematically investigated using, at least partly, visual tasks. The morphospace thus sheds some light on both strengths and weaknesses in our current understanding of cognition and represents an important step toward its improvement and conceptual advancement.

While the morphospace offers an intuitive grasp of the intricate dataset, it is not the endpoint of the exploration of the brain-cognition relationship. The morphospace properties have been exploited to evaluate the predictability of our knowledge of the underpinnings of cognition and how it affects subsequent investigations. Functions that are better investigated epistemologically follow a gradient from the centre to the periphery along the morphospace, exposing the limitations faced by fMRI studies in the past decades. In addition to sample

size[16,33], activation strength[34], and task design quality, the reliability of fMRI activations heavily depends on the specific function studied[35,36]. While some functions have shown test-retest reliability, consistency in the participation of brain structures, and clear psychological constructs[37–40], others have been poorly replicated due to weak experimental design and a lack of epistemological characterisation[41–43]. The study's findings reveal the reliably explored functions in the 30-year-old history of task-based fMRI literature. Our framework shows that maps located in the centre of the space have a low predictability index, suggesting low epistemological characterisation of their corresponding functions.

In contrast to task-related fMRI, resting state fMRI provides a useful reference point for assessing epistemological biases. According to the gradient theory, resting-state brain activation can be summarised based on the main trend of activation from unimodal to higher-cognition regions[22]. Our results indicate that brain regions with the most reliable functional activations are predominantly located within unimodal systems, with a few exceptions for monitoring, language, and attentional core regions (see Supplementary Materials for an extensive description). Moreover, our analysis of the left and right hemispheres' characterisation revealed a leftward asymmetry in the predictability index. This asymmetry may reflect variations in the coherence of the functional maps associated with different cognitive functions, suggesting that studies focusing on cognitive functions related to the left hemisphere yield results that are more consistently replicated across the literature. Some functions are more lateralised than others (e.g.[44,45]), and our results indicate that the right hemisphere has been less characterised due to limitations intrinsic to its exploration and merits prioritisation in future cognitive investigations using new tasks.

Such endeavour in exploring new functions is now feasible since our results demonstrate that new task-related activation maps can be reliably predicted to populate the gaps within and between the morphospace branches. The prediction of activation maps highlights the anatomical fluidity of the transition from one cognitive domain to another, suggesting that functions are not strictly segregated but rather fluidly expressed in the brain. Furthermore, the decoding of activation states from fMRI maps acquired during movie observation—an approach that is increasingly contributing to the exploration of the task-free engagement of the mind[46]—indicates that the morphospace can predict inner cognitive states, overcoming the limitations of task-based exploration.

Like studies focusing on task-based fMRI, the current study is challenged by methodological and theoretical limitations. The morphospace is based on meta-analytic activation maps built from heterogeneous activation studies, as thirty years of fMRI exploration entangle enormous methodological advances. However, in the context of Neurosynth coordinate-based meta-analytic analyses, the necessity for dataset normalisation is circumvented. This method uses coordinates rather than quantitative values and sidesteps the potential discrepancies that might arise due to variations in scanners or experimental designs. Furthermore, our framework, which uses studies from 1997 to 2017, has undergone rigorous validation. This included an expansion to encompass results from 2018 to 2021 that replicated findings ranging from 1997 to 2017 (see Supplementary Materials). Moreover, the robustness of our predictive framework has been tested with out-of-sample data benefiting from contemporary technological advancements (i.e., meta-analytic maps accounting for recent single findings and single activation results of papers up to 2022) and produced satisfactory results. This thorough examination underscores the generalisability of our predictive framework as it adeptly navigates the nuanced terrain of new datasets. Nevertheless, the predictive ability of the morphospace, expressed as the rationality index, raises the inverse error concern. In fact, the ability to predict activation patterns in a particular brain structure doesn't necessarily

imply a comprehensive understanding of its functions. While predictability does not equate to a full functional epistemological characterisation, we use it in the context of our study as a proxy measure of the degree of understanding of the brain-cognition organisation. Finally, to characterise the shape of the morphospace, we chose the neuron-shape metaphor. Metaphors can sometimes add an element of confusion. However, we chose the neuron metaphor to qualitatively describe the patterns observed in morphospace structure. This metaphor provides a useful way to refer to different parts of the structure, like branches and body, which are essential for navigating the morphospace properties.

In conclusion, the morphospace predictive framework and the predictability index can empower researchers to navigate the complexity of brain-cognition organisation comprehensively, offering a robust alternative to intuition-led explorations without constraining innovative enquiry. The predictability index, in particular, challenges and renegotiates the beliefs of the brain-cognition organisation on which fMRI studies have based their theoretical and experimental models for the past three decades. Future research has the potential to explore uncharted aspects of the brain cognition organisation with our framework, providing a systematic approach for hypothesis generation and experimental design. The morphospace, as conceptualised in our study, serves not only as a detailed map but also as a catalyst for refining and advancing the understanding of the brain's intricate workings.

## Methods

### Meta-analytic data thresholding and parcellation

Brain functions were derived from the open-access Neurosynth[24] database, an automated platform that computes whole-brain meta-analytic activation maps for specific cognitive terms. In short, Neurosynth applies an Automated Coordinate Extraction tool to retrieve the coordinates provided in the tables of activation papers. A term-mining approach identifies terms that are used with a high frequency within each paper, generating a list of terms that frequently occur in several fMRI studies. The coordinates associated with papers that report a specific term are then selected, and the meta-analysis compares the coordinates reported in papers with or without the term of interest. Thus, the meta-analytic maps indicate the strength of the activations associated with a specific term as a z-score, for each brain voxel. Although the tool aggregates all the contrasts reported in the study and does not distinguish between activations and deactivations[24], Yarkoni and colleagues showed that these factors modestly influenced the meta-analytic results[24]. The comparison of meta-analytic maps created by Neurosynth and manually curated maps created by the authors indicates that the automated approach produces results similar to those of manual curation[24]. Moreover, the authors investigated the occurrence of reported activation and deactivation coordinates in fMRI studies and found that deactivation is less often reported[24].

The present study utilises a meta-analytic dataset comprising 506 functions representative of the state-of-the-art in the exploration of the anatomy of brain cognition from 1997 to 2017. This dataset, manually curated and verified in a prior study[45], encompasses 11406 fMRI literature sources. To replicate the quantitative relationship between the maps, as outlined in the Supplementary Materials section dedicated to the replication of the morphospace architecture in the 2021 dataset, only the meta-analytic activation maps that were accessible in the 2021 Neurosynth collection were included. As a result, 84 meta-analytic maps that were not present in the 2021 collection were eliminated from the original dataset[45].

The 506 meta-analytic maps (Neurosynth 2017) were thresholded at $z = 3.4$ ($p = 0.000337$) to ensure the generalisability of the brain-cognition architecture to recent meta-analytic data. This threshold became a default parameter in the latest versions of Neurosynth (e.g., 2021).

Each map underwent a comprehensive cortical (MMP[47]) and subcortical (AAL3[48,49]) parcellation. We thus obtained a total of 440 brain parcels per meta-analytic map and extracted mean z scores for each parcel. The thresholding and parcellation were applied in FSL (fslmaths and fslstats, respectively; https://fsl.fmrib.ox.ac.uk/fsl/fslwiki/FSL).

To ensure the reliability of the findings across varying numbers and types of parcellation, the study's results were replicated using three uniform parcellation approaches containing 100, 400, and 800 parcels[50] as well as a random shuffle of the MMP and AAL3 parcels between and within meta-analytic maps. The uniform parcellations were generated by identifying voxels of the MNI152 based on their anatomical location, utilising x, y, and z coordinates, followed by clustering using Python k-clustering to divide the brain uniformly into 100, 400, and 800 clusters (https://github.com/chrisfoulon/BCBlib). This approach parcellated brain voxels based on proximity, disregarding any functional and anatomical connections between neighbouring voxels. To examine the robustness of the results, all four versions of the parcelled dataset (100, 400, 800 parcellations and random shuffle) underwent dimensionality reduction via UMAP, and the Euclidean distances between all points in the resulting embeddings were computed. The correlations between the morphospace Euclidean distances and the additional four spaces obtained from the additional parcellation approaches demonstrate the strong robustness of the neuron-shaped architecture (100 parcels $r = 0.51$, 400 parcels $r = 0.68$, 800 parcels $r = .54$, Supplementary Fig. S1). In contrast, no correlation was found between the morphospace and the random-parcellation space ($r = .01$, Supplementary Fig. S1).

## Spatial embedding

To reduce the dimensionality of the parcelled meta-analytic dataset, the Uniform Manifold Approximation and Projection (UMAP[25]) algorithm was employed.

UMAP is a non-linear, low-dimensional projection technique that retains the core structure of data while reducing its dimensionality in a lower embedding[25]. The UMAP Python library (umap-learn.readthedocs.io) was utilised, with the algorithm approximating the manifold estimation based on 15 neighbouring data points using the default 'n_neighbors' parameter[25]. This number was a good trade-off between finer manifolds of connected data points and the representation of the global structure[30]. The minimum distance parameter (min_dist) determines the minimum distance allowed between points in the low-dimensional embedding while respecting the connection between neighbours of the manifold and was set to default = 0.1 in the present analysis[25] (see Supplementary Fig. S2 for morphospace calculation using variable parameters). We utilised a linear Euclidean metric to compute distances in the morphospace, ensuring that these distances accurately reflect the level of similarities between data points. This approach allows us to maintain a proportional relationship between points distances and their similarities, facilitating the application of linear statistical methods, such as linear regression and Pearson's correlation, in subsequent steps of our analysis.

The selection of the appropriate number of dimensions was based on comparing the Mean Absolute Error (MAE) between measured and predicted maps, as described below. Specifically, we compared the MAE of the three-dimensional approach to those of the 2D, 4D, and 5D approaches using the Wilcoxon Signed Ranks test (Shapiro-Wilk test for normality: $p < .001$ for the three pairwise comparisons). The results indicated that the MAE of the three-dimensional approach was better than that of the 2D ($z = 4.37$; $p < .001$) and 5D ($z = 5.18$; $p < .001$; Supplementary Fig. S3A) approaches. However, there was no significant difference between the MAE of the 3D and 4D approaches ($z = −.48$; $p = .63$). We then tested the difference in predictability index (see below for details on the predictability index computation) between the two approaches and found that the index provided by the 3D

embedding was significantly higher than that of the 4D approach ($z = 3.29$; $p < .001$; Supplementary Fig. S3B). Based on these findings, we decided to use 3D embedding to create the morphospace. The embedding resulted in the clustering of maps with similar activation patterns and the spreading apart of different maps. The Python library Pickle was used to store the embedded transformation of the meta-analytic dataset as a Python object, which enabled the dimensionality reduction and projection of external data in the same embedded space (https://github.com/vale-pak/BCS/tree/BCS_computation[51]).

## Predictability index computation

We leveraged the linear spatial relationship among meta-analytic activation maps to examine the predictability of the function's anatomy and developed an index referred to as the predictability index. This index captures the underlying coherence of the brain-cognition knowledge summarised by the morphospace.

Coherence refers to the comparison of a novel observation against predictions based on a corpus of former, established knowledge. In this study, predictions have been computed via linear regression, which allows the unveiling of the underlying relationships between variables[52]. The established knowledge available on the brain cognition-organisation was retrieved as the linear spatial relationship between the maps, which reflects their similarity. The linear spatial relationship was determined by computing the shortest Euclidean distances between the data points embedded in the morphospace. To predict the activation pattern of each 506 map, we used voxel-wise linear regressions using FSL's Randomise tool. The goal was to create a model that could predict the activation pattern of a new target map based on its proximity to the reference maps. Our prediction method involved interpolating a target map's activation pattern by analysing the weighted contribution of reference maps. The weighting was determined by each reference map's Euclidean distance to the point of interest in the morphospace (i.e., the target map). Each map within the morphospace was treated as a target map in 506 regression models, with the Euclidean distances between each target map and the remaining 505 reference maps serving as independent variables. The activation pattern of the reference maps was the dependent variable.

The regression analysis produced t-statistic maps, which were then converted to z-statistic maps for further examination. To maintain consistency with our measured meta-analytic maps, we applied a $z = 3.4$ threshold to the z-maps.

To assess the accuracy of our predictions, we calculated a predictability index using Pearson's R (fslcc in FSL). This correlation measured the alignment between the predicted activation pattern and the actual observed pattern of each meta-analytic map.

To test whether the predictability index is associated with the number of studies aggregated in each meta-analytic map, we correlated the index obtained from an out-of-sample cohort of meta-analytic maps and their number of studies. Pearson's correlation coefficient ($r = 0.15$, Supplementary Fig. S4) indicates that the number of studies aggregated in each meta-analytic map is not a critical driver of the predictability index.

## Predictability index laterality

The parcels of the 506 predicted maps were divided into left and right hemisphere parcels, and the mean z-statistic of left and right structures was computed for each predicted functional map. Then, the t-test comparison was conducted in JASP to explore the mean predictability index differences between the left and right hemispheres.

## Predictability map computation and correlation with activation gradients

The predictability index computed for each of the 506 meta-analytic maps was used as an independent variable in a linear regression with the measured maps as dependent variables via randomise tool of FSL.

The regression allowed for identifying structures associated with a high predictability index. To test whether the predictability pattern of this map corresponded to a typical gradient activation, Spearman's correlations were computed between the obtained predictability map and each of the five activation gradients[22].

### Addressing the role of spatial autocorrelation

Recent studies have shown that the inherent spatial autocorrelation of brain maps can lead to spuriously high correlation, even between maps that are randomly generated. To address this issue, null maps with preserved spatial autocorrelation can be used as a reference point to assess the similarity between brain maps. In this context, if the findings of our study are led by spatial autocorrelation, null maps with preserved spatial autocorrelation should display the same space morphology and predictability index as the measured data.

Accordingly, we created null maps with preserved spatial autocorrelation using the procedure as in Markello and Misic[53] and Burt and colleagues[54]. Specifically, the 506 Neurosynth meta-analytic maps were inflated to a mid-grey projection of FreeSurfer's fsaverage5 surface using nearest neighbour interpolation. Each surface was then parcelled using the 400 parcels version of the Schaefer and colleagues' atlas[55], as processed and openly provided by the authors (https://github.com/netneurolab/markello_spatialnulls[54]). As a mid-step, we tested the robustness of the morphospace features with a new space computed using the newly processed and parcelled maps. Pearson's correlation of the Euclidean distances of the original and new morphospace indicates that the space can be replicated in the new processing and parcellation methods of the Neurosynth maps ($r = 0.78$, Supplementary Fig. S5A). After computing the geodesic distances between the 400 parcels as in ref. 53, we proceeded to the computation of the spatial null maps using the Burt and colleagues method[54]. The method randomly permutes the values in each surface-projected and parcelled map and smooths and re-scales the permuted values to reintroduce the spatial autocorrelation features of the original, non-permuted data. One thousand permutations were exploited to obtain a null distribution of surrogate maps for each Neurosynth map. A single surrogate map representing the measured Neurosynth map under the null distribution with preserved autocorrelation was predicted via linear regression from the 1000 surrogates of the null distribution. Supplementary Fig. S5B shows an example of the preservation of spatial autocorrelation of the 'auditory' empirical map in the corresponding surrogate map. We parcelled the spatial nulls using the Schaefer and colleagues' atlas[55], exploited the spatial nulls to build a 3D morphospace via UMAP, and tested the replicability of our original results in the random, autocorrelation-preserved dataset. We performed this comparison between the morphospace obtained from surface-projected and parcelled Neurosynth maps and the morphospace obtained from their surrogate. The low correlation coefficient ($r = 0.29$) between Euclidean distances of the empirical and spatial-nulls-derived morphospace suggests that the clusterisation of the maps is only partially driven by spatial autocorrelation (Supplementary Fig. S5C).

Finally, to validate the linear regression and correlation results used to build the predictability index, we tested the original predictability index and the one computed using predicted nulls. To compute the null-derived predictability index, we used the Euclidean distances between the surrogate maps to predict the morphology of the original 506 Neurosynth maps and measured the correlation between the resulting spatial-nulls-derived predictions and the original Neurosynth maps. The comparison between the original predictability index from our study and the predictability index obtained from the null surrogates shows that spatial autocorrelation alone does not contribute to the predictability degree of the maps ($r = 0.094$, Supplementary Fig. S5D).

### Embedding transformation and prediction of new functions

We utilised new and unexplored meta-analytic and raw-data functional activation maps to expand the utility of the morphospace and evaluate its predictive capacity (https://github.com/vale-pak/BCS/tree/New_maps_projection[51]).

We selected 888 meta-analytic maps referring to terms that were not part of the morphospace from the Neuroquery repository (https://neuroquery.org/). The terms referred exclusively to cognitive functions from the healthy adult brain.

We also obtained thirteen new cognitive terms and their corresponding meta-analytic maps from the 2021 version of Neurosynth (14371 literature sources), following the same exclusion criteria as the 2017 dataset[45].

In addition, we sought to include task-related activation maps from studies published after 2017, focusing on cognitive domains or functions identified as having high predictability indices. A total of nineteen new activation maps were carefully selected from Neurovault (https://neurovault.org/), excluding those from studies involving psychiatric or pathological cohorts.

These new maps were then projected onto the existing morphospace using the UMAP 'transform' tool. This process embeds new data into a pre-learned space without altering its structure. This embedding estimated the hypothetical locations of the new data within the morphospace based on their similarity to the 506 maps used to construct the space. Linear regressions were subsequently conducted to predict the anatomy of these new maps from their positions relative to the established 506 maps. The resulting predictions were transformed into z-maps, thresholded at $z = 3.4$, and parcelled. The predictions were then compared to the actual empirical data through Spearman's correlations, generating a predictability index that quantifies the morphospace's ability to accurately forecast the anatomical features of cognitive maps not previously described.

### Topic modelling of Neuroquery terms

Topic modelling was conducted by analysing the co-occurrence frequency of cognitive terms in the Neuroquery database using the term-frequency inverse document frequency (tf-idf) weighted by the pointwise mutual information index (PMI). The PMI measures the likelihood that two terms co-occur in a study relative to their independent occurrence. The PMI was calculated as the logarithmic ratio of the joint probability of each term pair's co-occurrence to their marginal probabilities, with only positive PMI (PPMI) values retained. To prevent taking the logarithm of zero, a value of zero was assigned to the PMI of term pairs with no co-occurrences. This produced a positive N-by-N term co-occurrence matrix, where N is the total number of terms. Hierarchical Ward clustering was applied to the co-occurrence matrix, with the squared Euclidean distance as the distance metric. Three cluster levels (12, 25, and 55 topics) were included to represent broad, medium, and finely-resolved domain resolutions across the literature, as there is no standard for selecting an optimal number of clusters. Although the 12 clusters summarise functions in broad domains and the 55 clusters over-represent certain domains, we focused on the medium cluster approach (25 clusters). Supplementary Figs. S5A, B shows the predictability index and the predictability index of the nearest five neighbours of Neuroquery maps falling in the 12 and 55 clusters.

### Cognitive decoding of brain activations during movie observation

Brain activation data from resting state and movie observation were obtained from the HCP 7 T dataset, with comprehensive details on data acquisition and processing as in ref. 30 (https://www.humanconnectome.org/hcp-protocols-ya-7t-imaging). The movie-watching (MOVIE) fMRI data were acquired using a gradient echo EPI sequence with a 1000 ms repetition time (TR), 22.2 ms echo time (TE), 45 deg flip angle,

field of view (FOV) of 208 × 208 mm, 130 × 130 matrix, spatial resolution of 1.6 mm3, 85 slices of 1.6 mm isotropic voxels, multiband factor of 5, image acceleration factor (iPAT) of 2, 7/8 partial Fourier sampling, echo spacing of 0.64 ms, and 1924 Hz/Px bandwidth, on a 7 Tesla Siemens Magnetom scanner. The MOVIE runs were acquired in 2 of the 4 total sessions, each lasting approximately 1.25 hours, starting with a 16-minute resting-state acquisition followed by 2 MOVIE runs. Participants were instructed to passively watch a series of video clips with audiovisual content, separated by 20 s of rest as indicated by the word "REST" in white text on a black background.

Clips from MOVIE1 and MOVIE3 runs were extracted from independent films that were openly available under Creative Commons licence on Vimeo, while MOVIE2 and MOVIE4 were clips from Hollywood films[30]. A total of 3655 t-stat activation maps were generated, with each MOVIE run comprising 4 to 5 video clips. MOVIE and rest runs 1–4 were 921, 918, 915, and 901 frames per run, respectively. To allow for comparison with the morphospace meta-analytic maps, all t-stats activation maps underwent z-transformation and were parcelled following the same multi-parcellation approach of the morphospace meta-analytic maps. The activation maps were thresholded at $z = 3.4$. The 3655 activation maps were then embedded into the 3D morphospace via UMAP, and the coordinates of the locations of each activation map were extracted from the morphospace. The Euclidean distances between each activation map and the 506 morphospace-measured maps were calculated, and the closest morphospace meta-analytic map to each activation map was identified. To match each movie frame and the activation map with the respective closest morphospace term, the hemodynamic delay was taken into account, including 5 s after frame offset.

### Reporting summary
Further information on research design is available in the Nature Portfolio Reporting Summary linked to this article.

## Data availability
The raw data and the data generated in this study are freely available at https://github.com/vale-pak/BCS.git[51]. Source data are provided in this paper.

## Code availability
The code created for the analyses conducted in the study is freely available at https://github.com/vale-pak/BCS.git[51].

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

## Acknowledgements

The authors would like to thank Dr Gaël Jobart, Dr Marc Joliot and the Groupe d'Imagerie Neurofonctionelle, and Pr Maurizio Corbetta and his team for the inputs and discussions on the study. This work was supported by the NextGenerationEU PNRR grant No. SOE_000013 (VP, EFFORT), the Marie Skłodowska-Curie grant agreement No. 101028551 (SJF, PERSONALISED), and the Donders Mohrmann Fellowship No. 2401515 (SJF, NEUROVARIABILITY), the European Research Council (ERC) Starting Grant agreement No. 757672 (DW, NeuroLag). M.T.d.S is supported by HORIZON- INFRA-2022 SERV (Grant No. 101147319) "EBRAINS 2.0: A Research Infrastructure to Advance Neuroscience and Brain Health", by the European Union's Horizon 2020 research and innovation programme under the European Research Council (ERC) Consolidator grant agreement No. 818521 (DISCONNECTOME), the University of Bordeaux's IdEx 'Investments for the Future' programme RRI 'IMPACT', and the IHU 'Precision & Global Vascular Brain Health Institute–VBHI' funded by the France 2030 initiative (ANR-23-IAHU-0001).

## Author contributions

V.P. and M.Td.S. contributed to the study conceptualisation, data curation, formal analysis, investigation, methodology, writing of the original draft, and project administration. V.N., L.T., M.A. and S.J.F. contributed to the study conceptualisation, formal analysis, methodology, and writing–review & editing. DW provided the methods and materials of the meta-analytic dataset Neuroquery and contributed to the writing–review & editing.

## Competing interests

The authors declare no competing interests.
