## [Peer Review File · Nature Communications]

The morphospace of the brain-cognition organisationEditorial Note: This manuscript has been previously reviewed at another journal that is not operating a transparent peer review scheme. This document only contains reviewer comments and rebuttal letters for versions considered at *Nature Communications*.

Reviewer #1 (Remarks to the Author):

Thanks for addressing all the points raised by me. I am satisfied with this revision and recommend acceptance.

Reviewer #2 (Remarks to the Author):

I appreciate some of the changes the authors made in their revision. However, I must say that I still struggle with several components of the paper. I still don't think that "rationality index" is an appropriate term to use (perhaps a "predictability index" would be closer to what the authors are actually measuring). I also still struggle with understanding the implications of the work and the connections to philosophy. Perhaps I'm just not the right audience for this paper, and so I'm happy for this manuscript to be published if the remaining reviewers are better able to appreciate the significance of this work.

Reviewer #3 (Remarks to the Author):

I read this manuscript and the associated rebuttal letter from previous reviews with great interest, and I am grateful for the opportunity to review it. The authors use non-linear dimensionality reduction on a large database of meta-analytic brain maps (neurosynth) to identify clusters of topics pertaining to spatially related maps. They validate their results using additional meta-analytic data (including from a separate database, neuroquery) as well as movie-watching data. They cast their results in terms of the "rationality" of the exploration of brain patterns in the literature.

Briefly: I find the results interesting, perhaps even more so than some of the other reviewers seemed to. The authors have clarified the manuscript and added validation analyses about their choice of dimensionality, which is certainly an improvement over the previous version. The movie-watching decoding is particularly compelling. However, I also have some substantial methodological concerns, which I believe the authors should address in order to demonstrate that their results are meaningful. I also have some objections about the authors' framing of some of their findings, which should more carefully distinguish facts from interpretations and conjectures.

My main objection, however, is about the appeal to rationality. This does not seem warranted, and I encourage the authors to abandon this way of framing their results when revising the manuscript, in the interest of clarity. I provide more details below, on each of these points, to explain how the authors could take action to address of them at revision.

Rationality

I have to admit that the re-casting of linear regression as rationality seems to be a red herring. It does not help the authors to convey their message: on the contrary, in my experience as a reader I found it distracting (and it seems that the other reviewers also experienced this). The authors themselves acknowledge in the rebuttal that their use of rationality is not a universally accepted one. The results of the study are just as interesting and informative without inventing new terminology by attributing a new meaning to terms that already have another, rather distinct meaning, especially since the new meaning seems to be well captured by existing terms, such as "consistency", "reliability", or "coherence". The authors themselves repeatedly use the phrase "reliability of functional characterisation" (eg page 5) to explain their results, which seems much more appropriate and straightforward, without the need to invoke philosophical terminology. For example, the authors state: "In line with this idea, linear regression models aim to uncover inherent relationships among variables, and the correspondence between linear regression prediction and real observation could serve as a quantifiable index of rationality". Presumably, a rational agent should choose a non-linear model over a linear one, whenever the additional predictive value warrants the extra model complexity, as per Ockham's razor. Epistemological considerations aside: wouldn't this correspondence between predicted and actual values be simply the R-squared measure of variance explained, or out-of-sample validation score (depending on

whether "prediction" is meant as synonymous with "regression", or as assessed in actual new data)? I genuinely struggle to see the need for a new name for such established concepts.

Overall, I find myself agreeing with Reviewer #2 (point 3) that the term rationality is used inappropriately and could lead to unwarranted inferences due to its normative implications. The changes to the manuscript do not, in my view, address this concern. For example, in responding to Reviewer #2, the authors claim "Our use of the term "Rationality Index" aimed to go beyond mere predictability to highlight the structural coherence and causality". However, no causality at all is involved, as far as I can tell, since only measures of similarity (correlation, Euclidean distance) are used, and no causal manipulations are involved. So the claim does not stand.

I recommend the authors to abandon the use of this "rationality" framework and replace it with a more accurate descriptor, such as "coherence", or "consistency", which the authors repeatedly use when explaining their results (e.g., "the theoretical and experimental coherence or consistency"; "consistent and reliable research findings"; "differential reliability and consistency"), and which are much less loaded in terms of implications of normativity/optimalty. I think that by using the loaded "rationality" term, the authors are doing themselves a disservice, by drawing the reader's attention away from the results themselves, and towards semantics. I would also like to emphasise that I don't think the authors will lose anything, in terms of the value of their study, by switching to a more appropriate term.

If the authors truly need to use new terms, then I believe it is imperative for them to explain, clearly and compellingly, how their new terms are different from existing ones such as consistency or reliability.

Methodological concerns

Are the consistent clusters just those whose maps aggregate a larger number of studies? I apologize if perhaps I missed it, but would the results change if the same number of studies were included for each term among the 506 maps?

Line 60: "In line with this idea, linear regression models aim to uncover inherent relationships among variables, and the correspondence between linear regression prediction and real observation could serve as a quantifiable index of rationality". The UMAP embedding at the core of this work is nonlinear; isn't this at odds with the linear definition of rationality espoused by the authors? Why not use PCA, if linearity matters? And why go back to linearity (correlation) for assessing similarity, after using a nonlinear embedding? This inconsistency is confusing, and a careful rationale should be provided.

Line 405: "Nineteen new activation maps were selected, which were parcellated and underwent three-dimensional embedding in the morphospace space using UMAP. The distances between each new map and the 506 morphospace measured maps were then utilised in linear regressions to predict their anatomy". I sincerely apologize, but I do not understand what was done here, procedurally. The authors are re-doing the embedding with the new maps, and then checking whether the new maps' positions in the new embedding is close to where they would have been, based on individually correlating them with the various neurosynth maps? It is entirely possible that here I simply misunderstand what the authors did, but I believe that this point calls for a more thorough explanation of all the steps involved, in the Methods.

"To compute the rationality index, we employed Pearson's R, which compared each measured meta-analytic map to the corresponding predicted z-map." One major problem with the use of correlation as a measure of similarity between brain maps is that recent work has abundantly shown that the inherent spatial autocorrelation of brain maps can, by itself, induce spuriously high correlation, even between maps that are actually random. In fact, this was explicitly demonstrated in neurosynth maps (Markello and Misic, 2021, NeuroImage). This means that high correlation between two maps may indicate "rationality", or may indicate that one or both of them have high spatial autocorrelation. The highly rational clusters simply be those whose maps have unusual levels of spatial autocorrelation. To ensure that their results are meaningful, the authors should

run additional analyses to exclude this possibility. Even if it turned out that the “rational” clusters are in fact driven by autocorrelation, this would still be interesting – but this possibility must be investigated. Methods exist for taking into account spatial autocorrelation in both cortex and subcortex (Markello and Misic, 2021 NeuroImage; Burt et al., 2020 NeuroImage)

Because of this issue, I agree that a null model is important, but simply randomly shuffling parcels is not enough, as it would lead to inflated false positives. What would the morphospace look like, if it were based on random maps with preserved spatial autocorrelation? If similar clusters were found, it would be a reason to infer that what the morphospace is capturing is not the cognitive aspect, but rather purely the spatial autocorrelation. Rejecting this null hypothesis would make the paper substantially stronger.

Other terminology/phrasing

Line 201: “We have developed a new model of the brain’s organisation of cognition, the morphospace”. This seems a bit grandiloquent, as a way of saying that the authors employed dimensionality reduction and clustering on an existing database.

Line 225: “better investigated epistemologically”. This suggests a value judgment over the quality of the studies that contribute to the neurosynth database, which is not an appropriate conclusion to draw. I recommend re-phrasing by simply stating that some terms have lower coherence of functional maps, without making implicit claims about the origin of this lower coherence. The explanation that the authors use in the Rebuttal letter seems much more appropriate: “the asymmetry indicates that studies focusing on cognitive functions associated with the left hemisphere provide more robust results.”. I recommend using this explanation instead of the current, somewhat misleading claim of one hemisphere being “better investigated”.

“right hemisphere has been less systematically explored and merits prioritisation”. I do not see where the authors find evidence for this. This statement suggests that authors of neuroimaging studies are not looking at the right hemisphere as often as the left, which is not something that the present manuscript allows to conclude (and seems unlikely, except for language studies). Rather, what the present manuscript shows is that right-hemisphere results occur less often. But whether this is because of lack of exploration, or simply because results are harder to find in the right hemisphere, are separate possibilities that the present study does not disambiguate, and the authors should not arbitrarily mark one of these possibilities as the true one.

Response to Reviewer #4: “when new activation studies base their exploration on biased, heterogeneous knowledge, their findings display a low rationality index”. This is a conjecture that the authors make, about the possible reasons underlying a low rationality index. However, there is currently no guarantee that this phenomenon will or has in fact occurred, as far as I can tell: the authors did not look into the individual studies that contributed to neurosynth to identify what knowledge each study’s authors were using to formulate their scientific hypotheses. To be clear: I am not saying that the authors should follow this procedure (though they may if they so choose). Rather, I am saying that the claim in question should be explicitly marked as a speculation rather than a fact, and throughout the manuscript, the authors should be more careful to disambiguate what they conjecture, versus what they actually know (as in, can prove to be true with data).

I believe that these points are addressable, and I hope that in doing so the authors will obtain a stronger and clearer manuscript.

Reviewer #1 (Remarks to the Author):

Thanks for addressing all the points raised by me. I am satisfied with this revision and recommend acceptance.

R: We would like to thank the reviewer for taking the time to review our manuscript and providing positive feedback on our revisions. We appreciate the reviewer's valuable insights that have helped us to enhance the quality of our work.

Reviewer #2 (Remarks to the Author):

I appreciate some of the changes the authors made in their revision. However, I must say that I still struggle with several components of the paper. I still don't think that "rationality index" is an appropriate term to use (perhaps a "predictability index" would be closer to what the authors are actually measuring). I also still struggle with understanding the implications of the work and the connections to philosophy. Perhaps I'm just not the right audience for this paper, and so I'm happy for this manuscript to be published if the remaining reviewers are better able to appreciate the significance of this work.

R: We thank the reviewer for the remarks. We have carefully considered their comments on the "rationality index" and have made the necessary revisions to the manuscript. As suggested by the reviewer, we have replaced the term with "predictability index" throughout the paper. Additionally, we have expanded the section on the study implications to provide a more comprehensive understanding of the impact of our findings.

All the changes have been highlighted in cyan in the manuscript. The edits to the Discussion section (page 13) have been also reported here:

“In conclusion, the morphospace predictive framework and the **predictability index can empower researchers to navigate the complexity of brain-cognition organisation comprehensively, offering a robust alternative to intuition-led explorations without constraining innovative inquiry.** The **predictability index**, in particular, challenges and renegotiates the beliefs of the brain-cognition organisation on which fMRI studies have based their theoretical and experimental models for the past three decades. **Future research has the potential to explore uncharted aspects of the brain cognition organisation with our framework, providing a systematic approach for hypothesis generation and experimental design.** The morphospace, as conceptualised in our study, serves not only as a detailed map but also as a catalyst for refining and advancing the understanding of the brain's intricate working.”

Reviewer #3 (Remarks to the Author):

I read this manuscript and the associated rebuttal letter from previous reviews with great interest, and I am grateful for the opportunity to review it. The authors use non-linear dimensionality reduction on a large database of meta-analytic brain maps (neurosynth) to identify clusters of topics pertaining to spatially related maps. They validate their results using additional meta-analytic data (including from a separate database, neuroquery) as well as movie-watching data.

They cast their results in terms of the “rationality” of the exploration of brain patterns in the literature.

Briefly: I find the results interesting, perhaps even more so than some of the other reviewers seemed to. The authors have clarified the manuscript and added validation analyses about their choice of dimensionality, which is certainly an improvement over the previous version. The movie-watching decoding is particularly compelling. However, I also have some substantial methodological concerns, which I believe the authors should address in order to demonstrate that their results are meaningful. I also have some objections about the authors’ framing of some of their findings, which should more carefully distinguish facts from interpretations and conjectures.

My main objection, however, is about the appeal to rationality. This does not seem warranted, and I encourage the authors to abandon this way of framing their results when revising the manuscript, in the interest of clarity. I provide more details below, on each of these points, to explain how the authors could take action to address of them at revision.

R: We thank the reviewer for the positive remarks on the study. We have addressed the reviewers’ concerns regarding the methodology and the concept of “rationality index” in the following points. All changes have been highlighted in cyan in the manuscript and also reported in the current document at the end of each response.

Rationality

I have to admit that the re-casting of linear regression as rationality seems to be a red herring. It does not help the authors to convey their message: on the contrary, in my experience as a reader I found it distracting (and it seems that the other reviewers also experienced this). The authors themselves acknowledge in the rebuttal that their use of rationality is not a universally accepted one. The results of the study are just as interesting and informative without inventing new terminology by attributing a new meaning to terms that already have another, rather distinct meaning, especially since the new meaning seems to be well captured by existing terms, such as “consistency”, “reliability”, or “coherence”. The authors themselves repeatedly use the phrase “reliability of functional characterisation” (eg page 5) to explain their results, which seems much more appropriate and straightforward, without the need to invoke philosophical terminology.

For example, the authors state: “In line with this idea, linear regression models aim to uncover inherent relationships among variables, and the correspondence between linear regression prediction and real observation could serve as a quantifiable index of rationality”. Presumably, a rational agent should choose a non-linear model over a linear one, whenever the additional predictive value warrants the extra model complexity, as per Ockham’s razor. Epistemological considerations aside: wouldn’t this correspondence between predicted and actual values be simply the R-squared measure of variance explained, or out-of-sample validation score (depending on whether “prediction” is meant as synonymous with “regression”, or as assessed in actual new data)? I genuinely struggle to see the need for a new name for such established concepts.

Overall, I find myself agreeing with Reviewer #2 (point 3) that the term rationality is used inappropriately and could lead to unwarranted inferences due to its normative implications. The

changes to the manuscript do not, in my view, address this concern. For example, in responding to Reviewer #2, the authors claim “Our use of the term "Rationality Index" aimed to go beyond mere predictability to highlight the structural coherence and causality”. However, no causality at all is involved, as far as I can tell, since only measures of similarity (correlation, Euclidean distance) are used, and no causal manipulations are involved. So the claim does not stand.

I recommend the authors to abandon the use of this “rationality” framework and replace it with a more accurate descriptor, such as “coherence”, or “consistency”, which the authors repeatedly use when explaining their results (e.g., “the theoretical and experimental coherence or consistency”; “consistent and reliable research findings”; “differential reliability and consistency”), and which are much less loaded in terms of implications of normativity/optimalty. I think that by using the loaded “rationality” term, the authors are doing themselves a disservice, by drawing the reader’s attention away from the results themselves, and towards semantics. I would also like to emphasise that I don’t think the authors will lose anything, in terms of the value of their study, by switching to a more appropriate term.

If the authors truly need to use new terms, then I believe it is imperative for them to explain, clearly and compellingly, how their new terms are different from existing ones such as consistency or reliability.

R: We appreciate the reviewer’s insightful comments and acknowledge the concerns raised regarding our use of the term “rationality”. Upon reflection, we recognise that the term may inadvertently lead to confusion or misinterpretation, detracting from the core findings of our research. In response to that feedback, we have revised our manuscript to replace the term “rationality” with “predictability”. We believe that “predictability” more accurately encapsulates the essence of our statistical analyses, specifically those involving linear regression, and aligns closely with the established concepts mentioned by the reviewer.

To address the reviewer critique, we have not only updated the terminology but also elaborated on the methodological underpinning of what we now refer to as the “predictability index”. This revision aims to clarify the basis of our analysis and its relevance to assessing the coherence in the organisation of brain functions. Specifically, we employ the morphospace’s spatial properties to gauge the predictability of functional maps. This involves comparing new observations against predictions derived from a robust body of coherent findings. By quantifying the discrepancy between these predictions and actual observations through Pearson’s correlation, we can refine our understanding of the predictability associated with each functional map.

Accordingly, we have detailed this process in the results section (page 5):

“To evaluate the coherence of the organization of brain functions, we leveraged the spatial properties of the morphospace. This statistical assessment involves comparing new observations with predictions based on a corpus of established, coherent findings³². The divergence between these predictions and the actual observation is quantified, offering a metric to gauge and potentially refine the predictability of future estimations¹⁴⁻¹⁶. This approach is expressed by the predictability index, which utilises linear regression to assess the degree to which the characteristics of each functional map can be anticipated based on its similarities with others measured through tridimensional Euclidean distance. Thus, the

predictability index is derived by quantifying the correlation between the predicted and observed maps reflecting the predictability power of each map. Meta-analytic maps contributing to the coherence of the organisation of brain functions had higher predictability indices, as depicted in Figure 1d.”

In making these changes, we aim to enhance the manuscript’s clarity and ensure that our terminology accurately reflects our analytical approach without invoking unnecessary philosophical implications. We believe these revisions will redirect the focus to our study’s results and their significance in the field.

We are grateful for the opportunity to clarify our methodology and terminology, and we hope that these amendments address the reviewer’s concerns adequately.

Methodological concerns

Are the consistent clusters just those whose maps aggregate a larger number of studies? I apologize if perhaps I missed it, but would the results change if the same number of studies were included for each term among the 506 maps?

R: We appreciate the reviewer’s inquiry regarding the methodological robustness of our study and the potential influence of the number of studies aggregated in each meta-analytic map on the predictability index. This question prompted us to undertake a thorough analysis to examine whether the predictability index is inherently biased by the volume of studies contributing to each map.

Due to the maps in our morphospace being derived from data up to the year 2017 and the inability to retrieve the exact number of studies for each map from that period, we opted for a contemporary and analogous dataset for this analysis. We used 888 meta-analytic maps from Neuroquery, chosen for their similar predictability pattern to the morphospace maps, to ensure the relevance and applicability of our findings. This approach allowed us to project new maps onto the morphospace without altering its established features, thereby assessing cognitive characterisation and predictability levels in a consistent manner.

As in the 2017 dataset, Neuroquery maps that were positioned at the periphery exhibited higher predictability indices compared to those centralized within the space (as illustrated in Figure 3a of the manuscript, reported here in RFigure 1a).

Crucially, when we correlated the number of studies aggregated in each map with its predictability index, we found only a low Pearson’s correlation coefficient ($r = 0.15$, RFigure 1b).

This outcome suggests that the predictability index is not driven by the number of studies aggregated in each meta-analytic map, thus reinforcing the integrity and validity of our predictability measure.

RFigure1. The predictability index of 888 Neuroquery maps and its association with the number of terms aggregated in each map. a) The 888 Neuroquery maps were projected onto the morphospace and clustered according to their similarity with the maps of the morphospace. Triangles indicate each new map’s coordinate in the morphospace, while transparent circles indicate the morphospace meta-analytic maps’ location. Maps with higher predictability index are located at the periphery of the space, while maps with low predictability are clustered at the centre. b) The low Pearson’s correlation ($r = 0.15$) between the predictability index (x-axis) and the number of terms aggregated in each map (y-axis) reveals that the index is not associated with the magnitude of data used for each meta-analysis. Datapoints are represented in blue, and the regression line is indicated in red. r : Pearson’s regression coefficient. Dim: dimension.

This additional analysis has been meticulously documented and integrated into the Methods section of our manuscript (page 16), ensuring that our methodology remains transparent, and our results, verifiable.

“ To test whether the predictability index is associated with the number of studies aggregated in each meta-analytic map, we correlated the index obtained from an out-of-sample cohort of meta-analytic maps and their number of studies. The Pearson’s correlation coefficient ($r = 0.15$, Supplementary Figure S4) indicates that the number of studies aggregated in each meta-analytic map is not a critical driver of the predictability index.”

Furthermore, RFigure 1b, detailing the relationship between the predictability index and the number of aggregated studies, has been included in the Supplementary Materials (Supplementary Figure S4 on pages 4 and 5), providing the reader with a comprehensive view of the analytical process and its outcomes.

Through this in-depth examination, we aim to address the reviewer’s concerns directly and affirm the methodological soundness of our study. We believe that our results stand robust against the critique raised and hope that this additional analysis further clarifies the independence of the predictability index from the volume of studies aggregated in each meta-analytic map.

Line 60: “In line with this idea, linear regression models aim to uncover inherent relationships among variables, and the correspondence between linear regression prediction and real

observation could serve as a quantifiable index of rationality’’. The UMAP embedding at the core of this work is nonlinear; isn’t this at odds with the linear definition of rationality espoused by the authors? Why not use PCA, if linearity matters? And why go back to linearity (correlation) for assessing similarity, after using a nonlinear embedding? This inconsistency is confusing, and a careful rationale should be provided.

R: We appreciate the reviewer’s critical observation concerning the methodological approach of our study, particularly the interplay between the nonlinear UMAP embedding and subsequent linear analyses. This critique offers us a good opportunity to clarify our rationale and the coherence of our methodological framework. While it is true that UMAP is inherently non-linear, serving to reduce the dimensionality of the data while preserving its local and global structure, our choice of this method was driven by its ability to cluster complex, high-dimensional data structures in a comprehensible three-dimensional space. The subsequent use of Euclidean distances within this morphospace is a deliberate choice, meant to quantify the level of similarity between data points in a manner that is both intuitive and consistent with traditional linear analyses. Our decision to employ linear regression and Pearson’s correlations for predicting the voxels activation and related z-scores, subsequently termed the ‘predictability index’, was made with the intention of providing a quantifiable measure that could be readily interpreted within the existing statistical framework. The linear Euclidean metric, despite the non-linear nature of the UMAP embedding, ensures that distances in the morphospace are proportional to the similarities between data points, allowing us to apply linear statistical approaches effectively in subsequent analyses.

To address the reviewer’s concern and improve the clarity of our explanation, we have revised the description of our methodological approach in the methods section (page 15), specifically regarding the use of Euclidean metrics.

“We utilised a linear Euclidean metric to compute distances in the morphospace, ensuring that these distances accurately reflect the level of similarities between data points. This approach allows us to maintain a proportional relationship between points distances and their similarities, facilitating the application of linear statistical methods, such as linear regression and Pearson’s correlation, in subsequent steps of our analysis.”

Line 405: “Nineteen new activation maps were selected, which were parcellated and underwent three-dimensional embedding in the morphospace space using UMAP. The distances between each new map and the 506 morphospace measured maps were then utilised in linear regressions to predict their anatomy’’. I sincerely apologize, but I do not understand what was done here, procedurally. The authors are re-doing the embedding with the new maps, and then checking whether the new maps’ positions in the new embedding is close to where they would have been, based on individually correlating them with the various neurosynth maps? It is entirely possible that here I simply misunderstand what the authors did, but I believe that this point calls for a more thorough explanation of all the steps involved, in the Methods.

R: We deeply regret the confusion caused by our initial description of the methodology involving the integration of new activation maps into our morphospace analysis. To clarify:

The process begins by incorporating nineteen newly selected activation maps into the pre-established morphospace, which was originally created using 506 maps from Neurosynth. This integration was achieved without altering the existing structure of the morphospace using the ‘transform’ feature of UMAP. This feature is specifically designed to embed new data points (in our case, new maps) into a

previously learned space, thereby estimating their locations within this space based on the model's understanding of the multidimensional structure.

Once these new maps were embedded into the morphospace, we calculated their Euclidean distances to the existing 506 Neurosynth maps. This step was crucial for understanding the relative position of each new map within the morphospace, reflecting its similarity to previously analysed maps. We then employed linear regression analyses to predict the anatomical features of these new maps based on their calculated distances to the known maps within the morphospace. The goal of this approach was to gauge how well the morphospace could predict the anatomy of cognitive functions not previously described, based on their hypothesised locations within this space.

The comparison between the predicted features (derived from the regression analyses) and the actual empirical data of the new maps provided us with a predictability index. This index served as a quantitative measure of the accuracy with which the morphospace could forecast the anatomical characteristics of new cognitive maps.

In response to your feedback, we have revised the description of this procedure in the methods section on page 18. The revised section now reads:

“We utilised new and unexplored meta-analytic and raw-data functional activation maps to expand the utility of the morphospace and evaluate its predictive capacity (https://github.com/valepak/BCS/tree/New_maps_projection).

We selected 888 meta-analytic maps referring to terms that were not part of the morphospace from the Neuroquery repository (<https://neuroquery.org/>). The terms referred exclusively to cognitive functions from the healthy, adult brain.

We also obtained thirteen new cognitive terms and their corresponding meta-analytic maps from the 2021 version of Neurosynth (14371 literature sources), following the same exclusion criteria as the 2017 dataset⁴⁹.

In addition, we sought to include task-related activation maps from studies published after 2017, focusing on cognitive domains or functions identified as having high predictability indices. A total of nineteen new activation maps were carefully selected from Neurovault (<https://neurovault.org/>), excluding those from studies involving psychiatric or pathological cohorts.

These new maps were then projected onto the existing morphospace using the UMAP ‘transform’ tool. This process embeds new data into a pre-learned space without altering its structure. This embedding estimated the hypothetical locations of the new data within the morphospace, based on their similarity to the 506 maps used to construct the space. Linear regressions were subsequently conducted to predict the anatomy of these new maps from their positions relative to the established 506 maps. The resulting predictions were transformed into z-maps, thresholded at $z=3.4$, and parcellated. The predictions were then compared to the actual empirical data through Spearman's correlations, generating a predictability index that quantifies the morphospace's ability to accurately forecast the anatomical features of cognitive maps not previously described.”

“To compute the rationality index, we employed Pearson's R, which compared each measured meta-analytic map to the corresponding predicted z-map.” One major problem with the use of

correlation as a measure of similarity between brain maps is that recent work has abundantly shown that the inherent spatial autocorrelation of brain maps can, by itself, induce spuriously high correlation, even between maps that are actually random. In fact, this was explicitly demonstrated in neurosynth maps (Markello and Misic, 2021, NeuroImage). This means that high correlation between two maps may indicate “rationality”, or may indicate that one or both of them have high spatial autocorrelation. The highly rational clusters simply be those whose maps have unusual levels of spatial autocorrelation. To ensure that their results are meaningful, the authors should run additional analyses to exclude this possibility. Even if it turned out that the ”rational” clusters are in fact driven by autocorrelation, this would still be interesting – but this possibility must be investigated. Methods exist for taking into account spatial autocorrelation in both cortex and subcortex (Markello and Misic, 2021 NeuroImage; Burt et al., 2020 NeuroImage)

Because of this issue, I agree that a null model is important, but simply randomly shuffling parcels is not enough, as it would lead to inflated false positives. What would the morphospace look like, if it were based on random maps with preserved spatial autocorrelation? If similar clusters were found, it would be a reason to infer that what the morphospace is capturing is not the cognitive aspect, but rather purely the spatial autocorrelation. Rejecting this null hypothesis would make the paper substantially stronger.

R: We thank the reviewer for raising this crucial point. We tested the Reviewer’s hypothesis thereby, if the results of our study reflect the maps’ spatial autocorrelation, null maps with preserved spatial autocorrelation will display the same space morphology and predictability index as the measured data. Following the reviewer’s suggestion, we created null maps with preserved spatial autocorrelation using the procedure described in Markello and Misic (2021) and Burt and colleagues (2020). Specifically, the 506 Neurosynth meta-analytic maps were inflated to a mid-grey projection of FreeSurfer’s fsaverage5 surface using nearest neighbour interpolation. Each surface was then parcelled using the 400 parcels version of the Schaefer et al (2018) atlas, as processed and openly provided by the authors (https://github.com/netneurolab/markello_spatialnulls, Markello and Misic, 2021). We have previously demonstrated the robustness of our results through different parcellation methods and resolutions (Supplementary Figure 2, also reported below). Thus, we limited the spatial null analysis to one parcellation atlas and the closer resolution (400 parcels) to our original parcellation method (440 parcels).

RFigure2. Replication of the morphospace in four additional parcellation approaches. Pearson's r of the comparison between each additional parcellation and the morphospace is indicated for the 100 (top left), 400 (top right), 800 (bottom left) uniform parcellation approaches, and a control randomly shuffled parcellation (i.e. null model, bottom right).

As a sanity check, we tested the robustness of the morphospace features with a new space computed using the newly processed and parcelled maps. Pearson's correlation of the Euclidean distances of the original and new morphospace indicates that the space can be replicated in the new processing and parcellation methods of the Neurosynth maps ($r = 0.78$, RFigure 3a).

After computing the geodesic distances between the 400 parcels as described in Markello and Misisic (2021), we proceeded to the computation of the spatial null maps using the Burt and colleagues method (Burt et al., 2020). The method randomly permutes the values in each map, and smooths and re-scales the permuted values to reintroduce the spatial autocorrelation features of the original, non-permuted data. One thousand permutations were exploited to obtain a null distribution of surrogate maps for each Neurosynth map. A single surrogate map representing the measured Neurosynth map under the null distribution with preserved autocorrelation was predicted via linear regression from the 1000 surrogates of the null distribution. For instance, RFigure 3b shows that spatial autocorrelation of the 'auditory' empirical map is preserved in the corresponding surrogate map.

We parcellated the spatial nulls using the Schaefer et al atlas (2018), exploited the parcelled spatial nulls to build a 3D morphospace via UMAP, and tested the replicability of our original results in the random, autocorrelation-preserved dataset. The low correlation coefficient (0.29) between Euclidean distances of the surface-projected and spatial-nulls-derived morphospace suggests that the clusterisation of the maps is only partially driven by spatial autocorrelation (RFigure 3c).

Finally, to validate the linear regression and correlation results used to build the predictability index, we tested the original predictability index and the one computed using predicted nulls. To compute the null-derived predictability index, we used the Euclidean distances between the surrogate maps to predict the morphology of the original 506 Neurosynth maps and measured the correlation between the resulting spatial-nulls-derived predictions and the original Neurosynth maps. The comparison between the original predictability index from our study and the predictability index obtained from the null surrogates shows that spatial autocorrelation alone does not contribute to the predictability degree of the maps ($r = 0.094$, RFigure 3 d).

RFigure 3. Robustness of results against a spatial autocorrelation-preserving null model. a) Pearson’s correlation of the comparison between the 400 parcels resolution of the Schaefer and colleagues atlas (2018) applied to the maps’ surface projection and the original morphospace. b) The plot shows the distribution similarity between the surface-projected empirical and SA-preserving null versions, an example meta-analytic map (auditory). c) Pearson’s r of the comparison of the Euclidean distances between the surface-projected empirical and SA-preserving null maps, both parcellated via Schaefer and colleagues’ atlas (2018). d) Pearson’s r of the comparison between predictability indices obtained from the predicted 506 meta-analytic maps and the predicted SA-preserving null maps. The axial slices next to the x and y axes represent the example version of the auditory map predicted from the original morphospace and the SA-preserving null space, respectively. Dim: dimension. SA: spatial autocorrelation.

We have now acknowledged the role of spatial autocorrelation in the Methods section on pages 16 and 17 and integrated RFigure 3 in the supplementary materials on page 5 as Supplementary Figure S5.

Recent studies have shown that the inherent spatial autocorrelation of brain maps can lead to spuriously high correlation, even between maps that are randomly generated. To address this issue, null maps with preserved spatial autocorrelation can be used as a reference point to assess the similarity between brain maps. In this context, if the findings of our study are led by spatial autocorrelation, null maps with preserved spatial autocorrelation should display the same space morphology and predictability index as the measured data.

Accordingly, we created null maps with preserved spatial autocorrelation using the procedure described in Markello and Misić (2021) and Burt and colleagues (2020). Specifically, the 506 Neurosynth meta-analytic maps were inflated to a mid-grey projection of FreeSurfer's fsaverage5 surface using nearest neighbour interpolation. Each surface was then parcelled using the 400 parcels version of the Schaefer et al (2018) atlas, as processed and openly provided by the authors (https://github.com/netneurolab/markello_spatialnulls, Markello and Misić, 2021). As a mid-step, we tested the robustness of the morphospace features with a new space computed using the newly processed and parcelled maps. Pearson's correlation of the Euclidean distances of the original and new morphospace indicates that the space can be replicated in the new processing and parcellation methods of the Neurosynth maps ($r = 0.78$, Supplementary Figure S5a). After computing the geodesic distances between the 400 parcels as described in Markello and Misić (2021), we proceeded to the computation of the spatial null maps using the Burt and colleagues method (Burt et al., 2020). The method randomly permutes the values in each surface-projected and parcelled map, and smooths and re-scales the permuted values to reintroduce the spatial autocorrelation features of the original, non-permuted data. One thousand permutations were exploited to obtain a null distribution of surrogate maps for each Neurosynth map. A single surrogate map representing the measured Neurosynth map under the null distribution with preserved autocorrelation was predicted via linear regression from the 1000 surrogates of the null distribution. Supplementary Figure S5b shows an example of the preservation of spatial autocorrelation of the 'auditory' empirical map in the corresponding surrogate map. We parcellated the spatial nulls using the Schaefer et al atlas (2018), exploited the spatial nulls to build a 3D morphospace via UMAP, and tested the replicability of our original results in the random, autocorrelation-preserved dataset. We performed this comparison between the morphospace obtained from surface-projected and parcelled Neurosynth maps and the morphospace obtained from their surrogate. The low correlation coefficient ($r=0.29$) between Euclidean distances of the empirical and spatial-nulls-derived morphospace suggests that the clusterisation of the maps is only partially driven by spatial autocorrelation (Supplementary Figure S5c).

Finally, to validate the linear regression and correlation results used to build the predictability index, we tested the original predictability index and the one computed using predicted nulls. To compute the null-derived predictability index, we used the Euclidean distances between the surrogate maps to predict

the morphology of the original 506 Neurosynth maps and measured the correlation between the resulting spatial-nulls-derived predictions and the original Neurosynth maps. The comparison between the original predictability index from our study and the predictability index obtained from the null surrogates shows that spatial autocorrelation alone does not contribute to the predictability degree of the maps ($r = 0.094$, Supplementary Figure S5d).“

Other terminology/phrasing

Line 201: “We have developed a new model of the brain’s organisation of cognition, the morphospace”. This seems a bit grandiloquent, as a way of saying that the authors employed dimensionality reduction and clustering on an existing database.

R: We have now dimmed the tone of the sentence.

“We introduced a novel framework for understanding the brain’s organisation of cognition, the morphospace, which was derived from an extensive meta-analytic database of task-related functional MRI (fMRI).”

Line 225: “better investigated epistemologically”. This suggests a value judgment over the quality of the studies that contribute to the neurosynth database, which is not an appropriate conclusion to draw. I recommend re-phrasing by simply stating that some terms have lower coherence of functional maps, without making implicit claims about the origin of this lower coherence. The explanation that the authors use in the Rebuttal letter seems much more appropriate: “the asymmetry indicates that studies focusing on cognitive functions associated with the left hemisphere provide more robust results.”. I recommend using this explanation instead of the current, somewhat misleading claim of one hemisphere being “better investigated”.

R: We appreciate your feedback and understand the importance of articulating our findings in a manner that does not imply a judgment on the quality of studies contributing to the Neurosynth database. Upon reflection, we understand that the phrase “better investigated epistemologically” could be misinterpreted as suggesting a qualitative assessment of the research within specific domains, which was not our intention.

To address this concern, we have revised the contentious statement to more accurately reflect our findings without attributing any value judgment. The reviewed sentence now reads

“This asymmetry may reflect variations in the coherence of the functional maps associated with different cognitive functions, suggesting that studies focusing on cognitive functions related to the left hemisphere yield results that are more consistently replicated across the literature”.

This adjustment aims to clarify that our observation of asymmetry in the predictability index across cognitive functions is based on the coherence and replicability of functional maps, rather than an assessment of research quality.

“right hemisphere has been less systematically explored and merits prioritisation”. I do not see where the authors find evidence for this. This statement suggests that authors of neuroimaging studies are not looking at the right hemisphere as often as the left, which is not something that the present manuscript allows to conclude (and seems unlikely, except for language studies). Rather, what the present manuscript shows is that right-hemisphere results occur less often. But whether this is because of lack of exploration, or simply because results are harder to find in the right hemisphere, are separate possibilities that the present study does not disambiguate, and the authors should not arbitrarily mark one of these possibilities as the true one.

R: We apologise for the confusion. By the referred sentence, we mean that the right hemisphere has not been explored as comprehensively as the left. We agree that results might be harder to find, and their scarcity is not due to the neglect of the scientific community but rather to the methodological and theoretical approach used in the investigation of right-hemisphere cognition. We have now changed the text with “right hemisphere has been less **characterised due to limitations intrinsic to its exploration and merits prioritisation**“.

Response to Reviewer #4: “when new activation studies base their exploration on biased, heterogeneous knowledge, their findings display a low rationality index”. This is a conjecture that the authors make, about the possible reasons underlying a low rationality index. However, there is currently no guarantee that this phenomenon will or has in fact occurred, as far as I can tell: the authors did not look into the individual studies that contributed to neurosynth to identify what knowledge each study’s authors were using to formulate their scientific hypotheses. To be clear: I am not saying that the authors should follow this procedure (though they may if they so choose). Rather, I am saying that the claim in question should be explicitly marked as a speculation rather than a fact, and throughout the manuscript, the authors should be more careful to disambiguate what they conjecture, versus what they actually know (as in, can prove to be true with data).

R: Thank you for giving us the opportunity to clarify the distinction between our empirical findings and the speculative interpretations we offer. We acknowledge that our discussion regarding the potential reasons underlying a low predictability index ventures into conjecture as it suggests that biases and heterogeneities in the knowledge base could influence these findings. We understand the importance of clearly demarcating such speculations from our data-driven conclusions throughout the manuscript. In response to this valuable critique, we have reviewed our manuscript to make sure that any speculative assertions are explicitly identified as such. This includes the statement concerning the impact of biased and heterogeneous knowledge on the predictability index. We now make it clear that these interpretations are hypotheses derived from our observations rather than direct outcomes of our analyses.

Our analysis, rooted in the UMAP-based spatial morphology and the predictability index derived from Neurosynth meta-analytic data, aimed to quantitatively assess the coherence of cognitive functions as represented in the existing literature. This assessment, as you correctly pointed out, does not extend to an examination of the individual studies’ hypothesis-formulation processes. Our intention was to provide a macroscopic view of the cognitive landscape as it is currently represented in neuroimaging literature, acknowledging the limitations and influences of researchers’ methodology and epistemological approaches on its understanding and representation.

We have revised the manuscript accordingly, making sure to maintain the integrity of our empirical findings while responsibly engaging with the broader implications of our work. We hope these adjustments adequately address the reviewer's concerns.

I believe that these points are addressable, and I hope that in doing so the authors will obtain a stronger and clearer manuscript.

R: We thank the reviewer for providing us with valuable insights that have helped us improve the quality of our study and our manuscript